# Exploring the Relationship Between Hypertension and Cerebral Microvascular Disease

**DOI:** 10.3390/diseases12110266

**Published:** 2024-10-23

**Authors:** Vasiliki Katsi, Andreas Mavroudis, Ioannis Liatakis, Manousiadis Konstantinos, Konstantinos Tsioufis

**Affiliations:** First Cardiology Clinic, School of Medicine, University of Athens, 11527 Athens, Greece; vkkatsi@yahoo.gr (V.K.); kmanousiadis@gmail.com (M.K.); ktsioufis@gmail.com (K.T.)

**Keywords:** hypertension, cerebral small vessel disease, microcirculation

## Abstract

Background/Objectives: Hypertension exerts negative effects on the vasculature representing a key risk factor for cardiovascular disorders, cerebral and Cerebral Small Vessel Disease (CSVD). Methods: An extensive research in the literature was implemented in order to elucidate the role of hypertension in the pathogenesis of CSVD. Results: Hypertension-mediated vascular dysfunction and chronic cerebral hypoperfusion are closely linked to CSVD. CSVD encompasses a wide range of lesions depicted on brain Magnetic Resonance Imaging (MRI) or Computed Tomography (CT) scans. The presenting symptoms and clinical course are highly variable, as a significant proportion of patients remain asymptomatic. Nevertheless, CSVD is associated with an increased risk of stroke, dementia and mobility disorders. Various randomised controlled trials have been implemented trying to shed light on the effect of vascular risk-modifying agents and lifestyle interventions on the prevention and treatment of small vessel disease. Conclusions: Hypertension has a pivotal role in the pathogenesis of CSVD. However, further research is required for a better understanding of the relationship between blood pressure levels and CSVD progression.

## 1. Introduction

Hypertension (HTN) affects more than 60% of individuals aged older than 65 years [1]. It remains a major risk factor for cardiovascular diseases (CVDs) and appears to be responsible for approximately 8.5 million deaths from ischaemic heart disease, renal and other vascular diseases worldwide [2]. Of note, cerebral functions appear to be early targets of HTN-induced organ damage [1]. Cerebral small vessel disease (CSVD) is a disorder primarily affecting small arteries, capillaries, arterioles and venules causing various lesions (white matter hyperintensities, microbleeds, lacunes and microinfarcts) that can be observed on brain imaging with Magnetic Resonance Imaging (MRI) or Computed Tomography (CT) scans. Although many people with CSVD remain asymptomatic, some of the aforementioned lesions can accumulate over time and appear to be associated with an increased risk of stroke, dementia, depression and mobility disorders [3]. To evaluate the current evidence and its potential limitations, we reviewed the literature on the association of HTN and CSVD in population-based prospective studies.

## 2. Description of the Microcirculation

Regulation of blood pressure (BP) is to a great extent mediated by resistance arteries. In small arteries and arterioles (with lumen diameters < 350 μm and <100 μm, respectively), peripheral resistance is approximately 50% of its total value. Minor alterations in arterial lumen diameter are associated with great changes in peripheral arterial resistance considering that resistance is inversely correlated with the fourth power of blood vessel radius [4]. Although there is no universally accepted definition, microcirculation encompasses the microvessels of systemic circulation with a diameter < 20 μm and therefore includes arterioles, post-capillary venules, capillaries and their sub-cellular constituents [5]. However, it is not entirely clear whether vessels defined as small arteries with a lumen diameter > 150 μm should also be included [6].

Microcirculation is responsible for the oxygen delivery from the red blood cells in the capillaries to the parenchymal cells and to the tissue cells contributing to the proper organ function, and therefore, microcirculation plays a significant role regarding the proper function of the cardiovascular system [5]. Microvasculature modulates the vascular tone, and this is achieved through the endothelial release of compounds, such as nitric oxide, reactive oxygen species and arachidonic acid metabolites. Microcirculation has been the focus of increasing research in recent years, and a range of conditions from reduced tissue maximal perfusion to impaired dilation of isolated arterioles has been set under the so-called term “microvascular dysfunction” [7].

## 3. Cerebral Microcirculation and HTN

It is known that cerebral blood flow remains almost unaltered within specific upper and lower blood pressure values in hypertensive patients as an adaptive mechanism aiming to protect the brain tissue for both hypoperfusion and hypertensive encephalopathy [8]. On the other hand, essential hypertension is primarily characterised by an increase in peripheral vascular resistance, which is mediated either by a reduction in both the lumen diameter and number of arteries or by an increase in their length [9]. These alterations in cerebral vasculature driven by elevated blood pressure have a major impact on cerebral circulation leading to the impairment of cerebral vasodilatation [9,10].

Cerebral vascular hypertrophy and remodelling exert both favourable and unfavourable outcomes. Firstly, the aforementioned vascular changes appear to protect the cerebral circulation, which is particularly susceptible to pressure damage, in HTN, by attenuating the increase in cerebral microvessel pressure [10]. Secondly, the increase in wall thickness and reduction in the vascular diameter leads to a virtual normalisation of the vascular wall stress [11]. Finally, all these alterations seem to contribute to the autoregulation of the cerebral blood flow [10]. Experimental studies have shown that myogenic response is enhanced, being an adaptive mechanism in HTN, and this process is partly mediated by the increase in intracellular Ca^2+^ in vascular smooth muscle cells (VSMCs) through upregulation of the 20-hydroxyeicosatetraenoic-acid (20-HETE)–short transient receptor potential channel 6 (TRPC6) pathway. Moreover, HTN alters not only the expression of epithelial sodium and other ion channels but also the activation of enzymes involved in the modulation of VSMC contractility. By these adaptive changes, intracranial blood volume is maintained within the normal range and cerebral microcirculation is protected from high pressure-induced damage [12].

On the other hand, these cerebrovascular alterations may lead to an impairment of vasodilatation as a response to stimuli, such as hypoxia, hypercapnia and endothelium-mediated vasodilatation [13]. Moreover, experimental data have shown that both structural and functional stages in the cerebral microcirculation seem to be to a great extent age-dependent [14]. The reduction in arterial distensibility may represent a characteristic of more advanced stages of essential HTN, whereas an inward remodelling of small cerebral arteries occurs mainly in earlier stages of the disease [15]. Hypertension-mediated vascular dysfunction leads to a reduction in cerebral blood flow and chronic cerebral hypoperfusion (CCH) [16]. Among several mechanisms involved in CCH, the activation of inflammatory cytokines contributes to blood brain–barrier (BBB) dysfunction and the induction of pyroptosis, a form of inflammatory cellular death combining characteristics of both cellular apoptosis and necrosis. Both microglia and astrocytes pyroptosis are involved in the pathogenesis of CCH [17]. Cerebral white matter demyelination, neurodegeneration, vascular dementia and atherosclerosis represent cerebrovascular disorders induced by CCH-mediated prolonged ischemia and reduced oxygen supply [17,18].

## 4. Endothelial Involvement and the Role of Oxidative Stress in HTN-Induced Cerebral Microvascular Disease

Brain functions such as tissue oxygenation, transport of metabolites and interstitial fluid balance are mediated by the endothelium of cerebral vessels through regulation of cerebral blood flow and interaction among endothelial cells, astrocytes, pericytes and oligodendroglial cells [19].

Formed by microvascular endothelial cells of cerebral capillaries at the brain and spinal cord, the BBB plays a critical role, as it controls the influx and efflux of biological substances and protects cerebral parenchyma acting as an obstacle to the entry of drugs and compounds into the central nervous system contributing to the homeostasis of the brain microenvironment [20].

The presence of reactive oxygen species (ROS) at high concentrations in the brain, mediated by high oxygen consumption, leads to higher vascular resistance through the increase in vasoconstriction and inhibition of NO signalling and bioavailability contributing to endothelial dysfunction and increased permeability of BBB [21,22]. NADPH oxidases (NOX) constitute the main ROS source in blood vessels [23]. Moreover, the elevated production of ROS in central autonomic regions, which are related to blood pressure control, contributes to neurohumoral changes that drive hypertension [24]. Glutathione (GSH) plays a key role as an antioxidant substance contributing to the protection of cells against ROS [22]. During ageing, an imbalance between cellular pro-oxidant and anti-oxidant mechanisms can occur, inducing oxidative stress, which has been implicated in many cerebrovascular diseases, including Alzheimer’s disease (AD), Parkinson’s disease (PD), amyotrophic lateral sclerosis (ALS), multiple sclerosis (MS), HIV-associated neurocognitive disorder, cerebral ischemia/reperfusion injury (I/R) and traumatic brain injury (TBI) [25].

## 5. Genetic Susceptibilities and CSVD

Apart from the established role of traditional risk factors, recent studies have shed light on the role of genetic factors in the pathogenesis of CSVD [26]. The best characterised monogenic form of CSVD is the cerebral autosomal dominant arteriopathy with subcortical infarcts and leukoencephalopathy (CADASIL), which is caused by mutations in the NOTCH3 gene on chromosome 19q12 [27]. Other rare monogenic forms of CSVD include CARASIL, CARASAL, COL4A1, COL4A2, RVCL-5, Fabry disease, pseudoxanthoma elasticum, HCHWA, Familiar British Dementia [27,28]. Sporadic forms of CSVD are neurological diseases, which are related to genetic variants and the presence of environmental factors, the strongest of which is hypertension [26]. The supportive features of a genetic aetiology in the onset and progression of CSVD include the presence of family history, age at onset younger than 50 y, absence of traditional vascular risk factors for ischemic stroke or cerebral haemorrhage, systemic findings and more severe manifestations on brain MRI [28].

## 6. Classification of CSVD

There are different types of small vessel diseases, and the following simplified classification has been proposed by the European small brain vascular disease expert group:Type 1: Arteriolosclerosis: This group includes lesions like fibrinoid necrosis, lipohyalinosis, microatheromas, microaneurysms and segmental arterial disorganisation. Loss of smooth muscle cells, the proliferation of fibroblasts, deposits of fibrohyaline material, degeneration of internal elastic lamina, narrowing of the lumen and thickening of the vessel wall are the main characteristics of this type. Other organs, such as kidneys and retinas, are also affected, and type 1 CSVD appears to be strongly linked with HTN. Effective blood pressure control seems to act favourably in the disease progression.Type 2: sporadic and hereditary cerebral amyloid angiopathy. In this group βA4 immuno-reactive, amyloid protein is progressively accumulated in small-to-medium-sized arteries and arterioles located mainly in the leptomeningeal space, the cortex and also in the capillaries and veins [29]. Cerebral amyloid angiopathy appears to be closely linked with Alzheimer’s disease; its frequency increases with age and, in some patients, is linked with large lobar haemorrhages [30]. It is also associated with microbleeds on neuroimaging and with the presence of cerebral ischaemic changes such as white matter lesions and microinfarcts.Type 3: inherited or genetic small vessel diseases distinct from cerebral amyloid angiopathy. This group includes conditions like Fabry disease, hereditary multi-infarct dementia of the Swedish type, MELAS (mitochondrial encephalopathywith lactic acidosis and stroke-like episodes), small vessel diseases caused by *COL4A1* mutations, hereditary cerebroretinal vasculopathy and CADASIL/CARASIL (cerebral autosomal dominant arteriopathy with subcortical infarcts and leukoencephalopathy/cerebral autosomal dominant arteriopathy with subcortical infarcts and ieukoencephalopathy, respectively).Type 4: inflammatory and immunologically mediated small vessel diseases, for example, Henoch–Schönlein purpura, Churg–Strauss syndrome, Wegener’s granulomatosis, microscopic polyangiitis, cryoglobulinaemic vasculitis, primary angiitis of the central nervous system and nervous system vasculitis associated with connective tissue disorders.Type 5: venous collagenosis.Type 6: other small vessel diseases such as post-radiation angiopathy.

The frequency of each type of CSVD is different, however, arteriolosclerosis and sporadic and hereditary cerebral amyloid angiopathy seem to be the most prevalent forms.

These types of CSVD are associated with adverse effects on brain parenchyma (neuronal apoptosis, diffuse axonal injury, demyelination and loss of oligodendrocytes) and a series of symptoms and unusual neuroimaging findings [29].

## 7. Imaging of CSVD

Historically, white matter lesions (WML) were the first imaging feature observed on CT scans of patients with CSVD and were described under the term “leukoaraiosis” [31]. With the widespread use of MRI, these lesions were termed white matter hyperintensities (WMH) due to their hyperintense appearance on T2-weighted MRI sequences [32]. MRI remains an important neuroimaging modality in CSVD, and the main neuroimaging features include small subcortical infarcts, white matter hyperintensities (WMH), cerebral microbleeds (CMBs) and prominent perivascular spaces (PVS) [33]. Cortical superficial siderosis (cSS) and cortical microinfacts represent also possible imaging features in MRI scans of patients with cerebral amyloid angiopathy [34].

Age and HTN represent the most important risk factors for WML progression [35]. Characterisation of WML severity is implemented either by visual grading or by quantification of WMH volume [33], with the latter being more reliable and more sensitive with the use of automated software [36]. WMH volume is quantified according to the pixel intensity of a determined area on T2-weighted MRI images [37]. Some pathological processes, like gliosis, edema and demyelination, lead to an increase in T2 signal intensity resulting in a “hyperintense” appearance compared to healthy tissue. As cerebrospinal fluid has also a bright appearance on T2-weighted imaging, the fluid-attenuated inversion recovery (FLAIR) sequence is used to remove the signal from CSF for better identification of parenchyma and pathological lesions [33].

On the other hand, lacunar infarcts of presumed vascular origin have been defined as subcortical, round or ovoid fluid-filled cavities with a diameter ranging between 3 mm and about 15 mm. Lacunar infarcts may be surrounded by a hyperintense rim, although this finding is not specific, as it can also exist around perivascular space. The minimal diameter of 3 mm has been used to define perivascular spaces, whereas the upper limit of 15 mm has been defined for the distinction between lacunar and larger subcortical infarcts [32]. Although HTN is the major risk factor for stroke, its role is crucial for the development of lacunar infarcts compared with large artery atherosclerosis and non-lacunar infarcts [38].

Perivascular spaces represent extensions of the extracerebral fluid that surround arteries and veins following vessels through the grey or white matter [32,39]. Perivascular spaces, as their diameter does not usually exceed 3 mm, are not visible on brain MRI imaging; however, larger spaces are more prominent in older patients at the base of the brain. With a diameter of less than 3 mm, especially when depicted perpendicular to brain vessels, perivascular spaces can be distinguished from lacunar infarcts [32].

Cerebral microbleeds (CMBs) represent focal areas of signal void on T2-weighted imaging with a diameter of up to 10 mm [32]. These lesions are related to previous microbleeding and hemosiderin deposition [40]. The location of CMBs is different according to aetiology. CMBs located at the cerebral cortex indicate amyloid angiopathy, whereas hypertensive angiopathy is linked with the formation of CMBs in the basal ganglia or thalamus [41]. Moreover, the use of susceptibility-weighted imaging (SWI) during MRI scans as a technique to enhance tissue contrast has led to the reduction in artefacts, better discrimination of brain tissues and increased detection of CMBs [42].

## 8. Biomarkers in CSCD

Apart from MRI, the role of biomarkers regarding the diagnosis and monitoring of CSVD is significant, and some of these indices are linked with specific types of CSVD. Apolipoprotein E (APOE) genotype, especially APOE-ε4, represents a genetic marker for sporadic cerebral amyloid angiopathy (CAA), but its use as a risk factor for Alzheimer’s disease (AD) has also been described, whereas the APOE-ε2 biomarker is predominantly expressed in subjects with CAA-associated intracerebral haemorrhage.

Both inflammatory (C-reactive protein, IL-1a, IL-6, homocysteine, VCAM-1 and von Willebrand factor) and coagulation markers (thrombomodulin, D-dimers, thrombin-antithrombin values) appear to be closely related to non-amyloid CSVD. These known biomarkers reflect the involvement of several mechanisms, such as endothelial and BBB dysfunction and genetic factors, showing the close relationship between biomarkers and CSVD. However, more clinical research needs to be implemented for the possible use of biomarkers in the diagnosis and monitoring of CSCD [43].

## 9. CSVD Lesions Progression/Clinical Implications

Although it is considered that CSVD lesions remain stable, several studies report that WMH, depicted in FLAIR imaging, may regress, remain stable and expand inside brain tissue or cavitate [3]. Arterial hypertension and ageing are factors considered to contribute to the progression of WMH lesions [44].

The presenting symptoms and clinical course of CSVD are to a great extent variable. A significant proportion of the affected people are asymptomatic; however, stroke, cognitive decline, depression, dementia, extrapyramidal symptoms and mobility difficulties are some of the main clinical manifestations of CSVD. The affected brain location, the extent and the type of pathological lesions in addition to age, environmental and genetic factors are considered the main contributors to the great clinical variability of CSVD. Comorbidities can also influence the clinical course and appear to be more prevalent in patients aged > 65 years, especially those with dementia. The most frequent comorbidity in patients with dementia is mixed Alzheimer’s disease and vascular dementia. Extensive research is required to better understand the mechanisms and the effects of vascular and dementia comorbidities on the clinical course of CSVD [3].

## 10. Pharmacological Interventions

Acethylcholinesterase inhibitors used for the treatment of Alzheimer’s disease have been tested in clinical trials for their potential effect on vascular impairment, giving mixed results considering the unrepresentative proportion of patients with small vessel disease. Anticoagulants (warfarin and DOACs) are effective in the prevention of first and recurrent stroke; however, it is unclear whether they have favourable effects on patients with small vessel disease. The positive impact of antiplatelet agents, such as cilostazol, dipyridamole and triflusal, on endothelial dysfunction and recurrence of ischemic infarcts has been described in clinical trials; however, the proportion of patients with small vessel disease lesions, apart from lacunar infarcts, was not clear. Experimental studies have shown that antioxidants, VEGF antibodies, dipyridamole, cilostazol and pentoxifylline seem to have a favourable effect on BBB permeability, whose integrity is accelerated in both WMH and lacunar stroke. However, clinical trials in humans are lacking. While inflammation has a role in the pathogenesis of small vessel disease, the potential positive effect of immunosuppressive agents has not been confirmed in clinical studies. It is also already known that statins reduce both first and recurrent stroke and contribute to the increase in endothelial NO production; however, their effect on endothelial function and small vessel disease progression is neutral. Clinical trials studying the effect of methylxanthines, such as pentoxifylline, have described a positive impact of these agents on dementia prevention; however, patients with small vessel disease were underrepresented [45].

Various randomised controlled trials have also tried to shed light on the effect of vascular risk-modifying agents and lifestyle modifications both on the prevention and treatment of CSVD. The PRESERVE trial included 62 hypertensive patients with confirmed symptomatic lacunar ischaemic stroke and WMH in MRI scans (mean age 69 years, trial duration 3 months). Participants were randomised in a 1:1 ratio to either standard or intensive blood pressure lowering treatment strategy with the primary endpoint being the change in whole-brain cerebral blood flow. There was no difference regarding cerebral blood flow between treatment groups, suggesting that an intensive treatment strategy for blood pressure control does not reduce cerebral blood flow in patients with small vessel disease [35]. The Secondary Prevention of Small Subcortical Stroke (SPS3) trial included 3020 patients with lacunar ischaemic stroke with a mean age of 63 years. Bleeding episodes and mortality rates were greater among patients treated with long-term dual vs. single antiplatelet therapy without a benefit regarding the reduction in recurrent stroke [3]. In the same trial, intensive blood pressure control led to a reduction in intracerebral haemorrhages, but no reduction in recurrent stroke or cognitive impairment was observed [3]. Finally, intensive blood pressure control seems to have no impact on the reduction in the incidence of dementia or mild cognitive decline compared with standard blood pressure treatment strategy [46]. Table 1 summarises the data from major clinical studies assessing pharmacological interventions in patients with CSVD.

## 11. Conclusions and Future Considerations

HTN elicits multiple negative effects on the vasculature. Studies suggest that HTN may have a pivotal role in the pathological mechanisms underlying CSVD. HTN-related changes (e.g., vascular lesions, hypoperfusion, oxidative stress) can occur progressively in cerebral small vasculature, leading to brain dysfunction in subjects with poor control of blood pressure; however, further research is essential to clarify the relationship between BP levels and progression of CSVD lesions.

Moreover, as the involvement of genetic susceptibilities in HTN-mediated small vessel disease lesions is possible, the elucidation of their role is important to understand variations in small vessel disease lesion development and symptom manifestation to help tailor the clinical management of hypertensive patients and, with careful monitoring and treatment of HTN, to decline the progression or prevent neurologic impairment in patients with CSVD.

## Figures and Tables

**Table 1 diseases-12-00266-t001:** Randomised controlled trials assessing pharmacological interventions in patients with CSVD [46,47,48,49,50].

Trial/Study Design	Intervention	Population (N/Characteristics)	Results
SPS3 (RCT)	Anti-hypertensives	3020, lacunar stroke	No reduction in stroke with intensive compared with guideline BP reduction
PRESERVE (RCT)	Anti-hypertensives	62, HTN, lacunar stroke, WMH	Intensive BP reduction did not appear to lower cerebral blood flow
SPRINT MIND (RCT)	Anti-hypertensives	9361, HTN, no history of stroke	Intensive BP control did not significantly reduce the risk of probable dementia
VITATOPS (RCT)—substudy	Statins	81, history of statin use before stroke	Slower WMH progression with statin use before stroke than no statin
ECLIPse (RCT)	ASA/cilostazol	203, lacunar stroke	Cilostazol decreased cerebral arterial pulsatility in patients with WMH

Abbreviations: SPS3: Secondary Prevention of Small Subcortical Stroke, RCT: randomised controlled trial, BP: blood pressure, HTN: hypertension, WMH: white matter hyperintensities, SPRINT MIND: Systolic Blood Pressure Intervention Trial Memory and Cognition in Decreased Hypertension, VITATOPS: VITAmins TO prevent stroke, ECLIPse: Effect of Cilostazol in Acute Lacunar Infarction Based on Pulsatility Index of Transcranial Doppler, ASA: acetylsalicylic acid.

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
