# Peer review of "Exploring the Relationship Between Hypertension and Cerebral Microvascular Disease"

_diseases, 2024, doi:10.3390/diseases12110266_

Round 1
Reviewer 1 Report
Comments and Suggestions for Authors
The review manuscript entitled “Exploring the Relationship between Hypertension and Cerebral Microvascular Disease by Vassiliki Katsi et al. discusses the detrimental effects of hypertension on the cerebral vasculature, particularly its role in the progression of cerebral small vessel disease. The review emphasizes the importance of monitoring and treating HTN to prevent neurological impairment in CSVD patients; and it suggests further research into the relationship between blood pressure and CSVD progression and highlights potential therapeutic targets. The manuscript is well written and structured.
However, the review, could be improved adding paragraphs related to: mention genetic susceptibilities, factors or mechanisms that may play a role in the Cerebral Microvascular Disease. In addition, oxidative stress or hypoperfusion mechanism is missing. Also, it is important to mention the way in which emerging therapies could be applied in the clinical practice.
Author Response
Thank you for your comments. We agree with the comments. For genetic susceptibilities we have added in the manuscript evidence regarding this issue at the ''genetic susceptibilities and CSVD'' part of the manuscript. We have also added evidence regarding hypoperfusion at the ''cerebral microcirculation and HTN'' part of the manuscript. Evidence regarding oxidative stress and its relationship with CSVD have been added at the ''endothelial involvement and the role of oxidative stress in HTN-induced cerebral microvascular disease'' part of the manuscript. Emerging therapies for CSVD have also been added and analyzed at the 1st paragraph of ''pharmacological interventions'' part of the manuscript.

Reviewer 2 Report
Comments and Suggestions for Authors
In this interesting Paper, the Authors described the association of hypertension and cerebral small vessel disease (CSVD) in population-based prospective studies.
The manuscript is well written, each paragraph is exhaustively presented.
I have one suggestion for the Authors.
At the end of the paragraph 8. "Implications for clinical practice", the Authors could mention the usefulness of speckle tracking echocardiography, an innovative noninvasive imaging technique that measures the myocardial deformation properties of all cardiac chambers and allows the clinicians to identify the subclinical myocardial dysfunction.
The relationship between blood pressure levels and CSVD progression may be further investigated by quantifying the degree of myocardial fibrosis, assessed by speckle tracking echocardiography. The lower the magnitude of myocardial strain parameters, the greater the degree of myocardial fibrosis, the higher the probability of systemic atherosclerosis and therefore CSVD. Recent evidence suggests that the impairment in myocardial strain parameters is a strong predictor of acute and/or chronic cerebrovascular disease. The Authors could mention the following references: PMID: 23902759 and PMID: 34525440.
Further studies should be performed for evaluating the relationship between myocardial strain parameters and CSVD.
Author Response
Thank you for pointing out this comment. We agree with the comment. However myocardial strain parameters are influenced by comorbidities other than hypertension. In the reference with PMID: 23902759, lower GLS was associated with subclinical brain disease, however, the participants in the study have additional comorbidities other than hypertension, that may influence GLS values.
In the reference with PMID: 34525440, the global left atrial strain was reduced in patients with acute ischemic stroke in the Emergency Department, however we believe that we should emphasize to the relationship of hypertension with CSVD lesions, including patients with subclinical hypertension-mediated CSVD lesions.